# *Aspergillus* Endophthalmitis: Epidemiology, Pathobiology, and Current Treatments

**DOI:** 10.3390/jof8070656

**Published:** 2022-06-22

**Authors:** Alisha Khambati, Robert Emery Wright, Susmita Das, Shirisha Pasula, Alejandro Sepulveda, Francis Hernandez, Mamta Kanwar, Pranatharthi Chandrasekar, Ashok Kumar

**Affiliations:** 1Department of Ophthalmology, Visual and Anatomical Sciences, School of Medicine, Wayne State University, Detroit, MI 48201, USA; alisha.khambati@med.wayne.edu (A.K.); rwrigh@med.wayne.edu (R.E.W.III); susmita05@wayne.edu (S.D.); kanwarmamta@gmail.com (M.K.); 2Division of Infectious Diseases, Department of Internal Medicine, School of Medicine, Wayne State University, Detroit, MI 48201, USA; spasula@med.wayne.edu (S.P.); pchandrasekar@med.wayne.edu (P.C.); 3Wayne State University, Detroit, MI 48201, USA; alejandro.ponce@wayne.edu (A.S.); gg7926@wayne.edu (F.H.)

**Keywords:** fungal endophthalmitis, *Aspergillus*, pathogenesis, antifungal, diagnosis

## Abstract

Fungal endophthalmitis is one of the leading causes of vision loss worldwide. Post-operative and traumatic injuries are major contributing factors resulting in ocular fungal infections in healthy and, more importantly, immunocompromised individuals. Among the fungal pathogens, the *Aspergillus* species, *Aspergillus fumigatus*, continues to be more prevalent in fungal endophthalmitis patients. However, due to overlapping clinical symptoms with other endophthalmitis etiology, fungal endophthalmitis pose a challenge in its diagnosis and treatment. Hence, it is critical to understand its pathobiology to develop and deploy proper therapeutic interventions for combating *Aspergillus* infections. This review highlights the different modes of *Aspergillus* transmission and the host immune response during endophthalmitis. Additionally, we discuss recent advancements in the diagnosis of fungal endophthalmitis. Finally, we comprehensively summarize various antifungal regimens and surgical options for the treatment of *Aspergillus* endophthalmitis.

## 1. Introduction

*Aspergillus* belongs to a group of conidial fungi comprised of multiple species, including *A. fumigatus* [1]. The *Aspergillus* genus typically lives and propagates in decaying organic material or detritus and can commonly be found in areas inhabited by people, i.e., homes and hospitals with moist conditions [2,3,4]. Although *A. fumigatus* is the most prevalent species responsible for ~90% of human infections, *non fumigatus* species exposure is also being reported as causing fungal infections [1,5].

The most common complication caused by *A*. *fumigatus* is a hypersensitive reaction known as allergic bronchopulmonary aspergillosis. While fungal infections are less common, immunocompromised patients are at higher risk for fungal pulmonary infections, in which patients with cavities can develop formations of an Aspergilloma, a concentrated area of fungal growth in the lungs [6,7]. Although the respiratory tract is the main route of *Aspergillus* entry and site of colonization, the infection can rarely transmit to various parts of the body including the skin, kidneys, bones, and eyes [8]. Additionally, *Aspergillus* can directly invade the eye following injury or post-operative trauma resulting in endophthalmitis (Figure 1), resulting in increased inflammation, fungal load, and retinal tissue damage, and reduced retinal function. Once inside the eye, it evokes an inflammatory response that can ultimately lead to permanent blindness if not treated promptly [9]. However, across all microbes, fungal infection is responsible for only 5–10% of endophthalmitis cases [10]. A recent study by Joseph, J. et al., from India, reported that the *Aspergillus* spp. is the main causative agent of fungal endophthalmitis [10]. Although the prevalence (72% in 1991 compared to 27.3% in 2015) of *Aspergillus* endophthalmitis has greatly decreased, it continues to be one of the major threats to vision loss [10]. Gniadek et al. identified *Aspergillus* to be one of the most commonly found fungi in operating rooms, resulting in postoperative infections [4]. Although postoperative endophthalmitis typically has an acute onset within seven days following surgery, chronic postoperative endophthalmitis is less common, can occur weeks after surgery, and is caused by lesser virulent fungi [9].

In this review, we comprehensively discuss the possible causes of *Aspergillus* endophthalmitis and its pathobiology and host immune responses, laboratory diagnostic, and current antifungal treatment options.

## 2. Epidemiology

*Aspergillus* endophthalmitis can occur either through endogenous or exogenous routes [11,12]. While endogenous endophthalmitis is acquired hematogenously, the latter occurs due to ocular trauma or post-surgical complications [13]. In general, molds such as *Aspergillus* have been reported as one of the most common infectious agents for exogenous endophthalmitis [14], usually due to trauma and varying types of intraocular surgery [13,15,16,17,18,19,20]. Kernt et al. reported that 90% of the post-operative endophthalmitis cases occur after cataract surgery [9], in addition to an increased risk for Aspergillosis in post-cardiac surgery and organ transplantation patients [16]. However, most instances of fungal endophthalmitis are linked with intravenous drug use, an immunocompromised status, and prior intraocular surgeries, with rare reports for healthy individuals [20,21,22,23]. These nuances in *Aspergillus* presentation provide challenges in diagnostic and therapeutic regimens [24].

The incidence of endogenous endophthalmitis ranges from 2–15%, including those caused by bacterial and fungal pathogens [25], where *Aspergillus* is the second most common cause of endogenous endophthalmitis, with the primary fungal pathogen being *Candida albicans* [26,27,28,29]. Past medical history of chronic steroid use, chemotherapy treatment, and organ and bone marrow transplantation suggest an association between immunosuppression and disseminated *Aspergillus* infections [30], whereas other conditions, including cancer, diabetes mellitus, HIV, renal insufficiency, and lung disease, have been found to predispose individuals to *Aspergillus* endophthalmitis [18]. Hence, the presence of existing comorbidities and systemic diseases can result in *Aspergillus* endophthalmitis [12]. Autoimmune diseases, such as granulomatosis with polyangiitis, affect the eye across 50 to 60% of patients, and thus, can cause secondary endogenous endophthalmitis [26]. The degree of ocular involvement, such as from subretinal to choroidal infections, and leukopenia can also impact the severity of *Aspergillus* endophthalmitis [16]. The prevalence of *Aspergillus* endophthalmitis is reported as being around 7.1% among orthotopic liver transplant recipients [31], suggesting an increased susceptibility to *Aspergillus* in patients with organ transplants [31]. Another study reported an acute case of *Aspergillus* endophthalmitis to be related to aortic valve replacement in a healthy individual [17].

## 3. *Aspergillus* Species and Endophthalmitis

The *Aspergillus* species are ubiquitous saprophytic molds and act primarily as opportunistic pathogens in humans [32]. Ocular diseases caused by *Aspergillus*, such as endophthalmitis, are usually caused by the exogenous route of infection, which can occur via the contamination of surgical instruments with a higher incidence among immunocompromised patients [33]. In some cases, endogenous transmission also occurs when fungi gain access into the bloodstream from its primary site of infection (e.g., pneumonia), finally reaching the eye. There are numerous species of *Aspergillus* associated with human diseases, among which *A. fumigatus* and *A. flavus* cause the majority of fungal endophthalmitis cases [17,34,35]. Other species reported to cause endophthalmitis include *A. terreus*, *A. glaucus, A. ustus, A. terreus*, and *A. versicolor* [34]. Though rarely reported as pathogenic, *A. niger* has been shown to cause several diseases including otomycosis, cutaneous infections, pulmonary diseases, and endophthalmitis in humans [35,36,37]. *A. nidulans* has rarely been found to cause exogenous endophthalmitis, but recent reports suggest its involvement in causing endogenous endophthalmitis [38,39,40].

*a*.
*Aspergillus fumigatus*


*A. fumigatus* is reported worldwide as being present in the air at relatively higher concentrations and continuously inhaled [8]. It rarely results in any adverse effect, as it is effectively cleared by the host’s innate immune system. However, if *A. fumigatus* can evade the host’s immune response and begin to invade and replicate in the surrounding tissues, it can become life threatening [41,42,43]. Presently, its pathogenesis is associated with various virulent proteins promoting conidial growth or morphology and conferring resistance to the host’s antifungal mechanisms. Epithelial and endothelial host cells play crucial roles in internalizing the conidia through the Dectin-1-dependent pathway [44,45,46]. However, many specific host molecular defense mechanisms are still poorly understood and require further research. One such defense pathway is the recognition of the pathogen and subsequent activation of neutrophils that help with microbial killing. However, in non-neutropenic immunosuppressed models such as in genetic Chronic Granulomatous Disease (CGD) or corticosteroid-induced immunosuppressive conditions, the *Aspergilli* conidia persists inside the host due to an compromised fungicidal ability [47]. Additionally, *A. fumigatus* has been reported for its cellular heterogeneity, in terms of its virulence and ability to avoid host antifungal pathways, and hence requires further investigation for its possible role in drug resistance [43,46,48]. The first case of exogenous endophthalmitis by *A. fumigatus* was reported in a 68-year-old diabetic man with necrotizing scleritis following a vitrectomy [48]. Similarly, an immunocompetent 78-year-old Caucasian female was reported to develop *A. fumigatus* endogenous endophthalmitis [49].

*b*.
*Aspergillus flavus*


*A. flavus* is the predominant species of *Aspergillus* in Asia, the Middle East, and Africa, which can be attributed to its ability to endure hot and dry climates. *A. flavus* induces similar clinical syndromes as the other *Aspergillus* species, especially during sino-orbital aspergillosis and ocular infections [50,51,52,53]. A recent study of Indian ICUs reported that *A. flavus* is the common fungus isolated from non-classical risk-factor patients, with chronic obstructive pulmonary disease, diabetes, liver disease, and glucocorticoid use contributing to 63.5%. This surpassed patients with classical risk factors, with 36.4% of cases related to neutropenia, malignancy, and transplant recipients on immunosuppression therapy. On the other hand, studies of healthy and immunocompromised mice have also reported *A. flavus* to be more virulent, resulting in higher mortality [52,54,55]. The fungi have also been reported in patients with endogenous endophthalmitis [33,56]. Although there is very little literature on *A. flavus* being responsible for endogenous *Aspergillus* endophthalmitis in immunocompetent individuals, this is well recognized for immunocompromised individuals. A clinical study showed a 23-year-old female presented with a history of taking steroids for the past year developing *A. flavus* endophthalmitis that led to sudden vision loss [57]. About 34% of trauma cases were associated with *A. flavus* endophthalmitis [13]. Post-operative endophthalmitis has also been reported with this species [54]. *A. flavus* is rarely found to be resistant to itraconazole and voriconazole, but it appears to be frequently resistant to amphotericin B as compared to *A. fumigatus* [50,55].

## 4. Clinical Manifestations of Endophthalmitis

Clinically, *Aspergillus* endophthalmitis presents with symptoms of ocular pain and blurry vision, which can be seen in both endogenous and exogenous forms [30]. Additionally, photophobia can occur in individuals without any ocular trauma or in those who use contact lenses [21]. Unlike exogenous endophthalmitis, endogenous endophthalmitis can occur in both eyes, and its onset is typically subacute [25]. Chakrabarti et al. showed that out of 113 cases of both exogenous and endogenous fungal endophthalmitis, those with endogenous had visual acuity that was worse than their hand motions [58]. Additionally, since endogenous endophthalmitis often involves systemic spread, symptoms such as fever might prevail in the patients [59].

A decrease in visual acuity is also commonly seen in patients suffering from mycotic eye infections [17,48]. A clinical case report of a 76-year-old woman with a history of prosthetic aortic valve replacement was found to have right-sided ocular pain and reduced visual acuity [17]. This patient presented with loss of the red reflex, a dense vitreous, and prominent corneal opacification [17]. Other clinical presentations include keratic precipitates, hypopyon formation, and floating anterior chamber cells in the aqueous humor of the eye [19]. Hypopyons, found in 75% of symptomatic patients, are located posteriorly, and are commonly associated with creamy yellow exudates [60]. In addition, a study reported the presence of sub-macular abscesses as well (Figure 2) [61].

An important sign of *Aspergillus* endophthalmitis is the presence of inflammation in the affected eye interiors, including vitritis [9,63]. Chorioretinal infiltrates may also be present, with reports of prominent vitreous inflammation [19,63]. In a case report, Fincher et al. described a patient with left-sided redness and eye pain, who had idiopathic diffused anterior scleritis [24]. A study has reported retinal abscesses in an infected patient’s eyes, who was diagnosed with bilateral endogenous endophthalmitis. The 26-year old patient had a clinical presentation of dense vitreous exudates with bilateral subretinal abscesses located at the posterior pole of each eye [64].

Aspergillosis can cause choroidal and retinal vessel invasion, which is a distinct feature not found in *Candida* endophthalmitis [16]. Moreover, *Aspergillus* can lead to painful choroidal infiltration and lesions, differentiating it from protozoal and viral etiologies [65]. Hence, the anatomical features of *Aspergillus* endophthalmitis include a thickened choroid, lack of retinal tissue organization, hypopyon, and high prevalence of subretinal exudates (Figure 3) [65].

## 5. Pathobiology and Immune Response in *Aspergillus* Infection

Various signaling pathways are involved in the process of fungal recognition, followed by cytokine secretion, ultimately leading to cell-intrinsic pathogen killing. *Aspergillus* conidial development initiates inflammatory responses via the morphotype-specific exposure of β-glucans, activating the C-type lectin receptor dectin-1 [66]. These spleen tyrosine kinase-coupled receptors trigger the caspase recruitment domain-containing protein 9 (CARD9), forming a trimeric complex with BCL10 and MALT1 and leading to the induction of the NF-κB pathway and the production of proinflammatory mediators, such as *IL-1β*, *IL-6*, *IL12*, *CXCL1*, *CXCL2*, and TNF-a [67].

The neutrophil infiltration to the site of infection is the major innate host defense carried out by the dectin-1 and CARD9 signaling, post-aspergillosis [68,69]. Individuals suffering from Mendelian defects in *Clec7a* are usually not affected by aspergillosis, but in the case of allogeneic HSCT, *Clec7a* polymorphisms (in both donor and recipient genomes) potentiate the chance of pulmonary fungal infection [70]. In other words, dectin-1 signaling leads to increased *Aspergillus* susceptibility under the pretext of immune system damage.

In the lung, epithelial IL-1/MyD88 receptor signaling mediates early secretion of the neutrophil chemokine CXCL1 and the initial transport of neutrophils to infected airways [71] and activation of the MyD88 pathway, which is necessary for the neutrophil-mediated killing of conidia. IL-1α and IL-1β are also involved in this process [72,73]. Their relative effects can vary and are dependent on the *Aspergillus* strain, morphological types, and use of exogenous immunosuppression mice models. In later stages of infection, the hematopoietic CARD9 signal amplifies the chemotactic signal and maintains lung neutrophil recruitment. In humans, Mendelian defects in CARD9 can cause spontaneous extrapulmonary aspergillosis, and the resulting fungal abscess in affected patients does not contain neutrophils [74]. The absence of lung disease described in affected individuals suggests that the compensation pathway through IL-1R/MyD88 signaling in mice can adequately compensate for the loss of CARD9-dependent pulmonary neutrophil recruitment [74].

In the murine CGD model, IL-1 receptor signaling can lead to dysregulation of the inflammatory response, which is characterized by an excessive recruitment of neutrophils, antifungal activity, and changes in tissue hypoxia. In this case, anakinra (a recombinant IL-1 receptor antagonist) can moderately improve the survival rate of mice [75]. Anakinra treatment did not improve the disease severity in CXCR2^-/-^ mice, in which the recruitment of lung neutrophils was delayed. These findings emphasize the importance of intervention against the underlying mechanisms of self-inflicted immune damage [76].

The soluble pentraxin-3 aggregate protein covers the conidia of *A. fumigatus* in the alveolar space and is partially enhanced by a complement deposition, the FcγIIA receptor (CD32), and potentiates the complement receptor 3 (CD32)-dependent phagocytosis (CD11b/CD18) [77]. Another study showed pentraxin-3 induces interferon-β signaling through TLR4/MD-2 and an adaptor containing a TIR domain [78]. Md2^-/-^ mice have defects in the uptake and death of neutrophil conidia in the body. Consistent with these observations, the human PTX3 gene polymorphism is found to be associated with differences in the uptake of conidia by human neutrophils and the risk of invasive aspergillosis in HSCT patients [79].

Antibodies to or genetic defects in the β2 integrin subunit of complement receptor 3 (i.e., CD18) inhibit the activation of NADPH oxidase in neutrophils during *A. fumigatus* infection [80]. The antifungal signal transduction pathway may include the β and δ subtypes of phosphatidylinositol-3-kinase (PI3K) and protein kinase B (PKB, Akt), and is not affected by the deletion of CLEC7A or CARD9 genes [81]. However, in mice exposed to *A. fumigatus* conidia through the respiratory route, hematopoietic cells in the lung lacking CD18 are not associated with the inability to kill conidia via neutrophils [68]. This shows that the loss of CD18 in this model could be overcome by employing other cellular response mechanisms. In contrast, in the fungal keratitis model, the transport of CD18-dependent neutrophils to the cornea is essential for the defense of the host [82].

Patients with hyper IgE syndrome (i.e., Job syndrome; STAT3 deficiency) are susceptible to aspergillosis, usually at the age of 30 or 40 for those with bronchiectasis or emphysema [83]. Stat3-deficient human neutrophils can still inhibit the growth of swollen and germinated conidia of *A. fumigatus* and have normal chemotactic activity [84]. These patients are more susceptible to invasive aspergillosis due to recurrent bacterial infections, and the model of this anatomical lung defect reflects this. Ruxolitinib, a small JAK/STAT inhibitor molecule, can reduce the death of human neutrophils stimulated by IL-6 and IL23 during *Aspergillus* infection [85]. These data support the hypothesis that JAK/STAT signaling contributes to the neutrophil-mediated clearance of *Aspergillus*, although the exact mechanism has not yet been determined [86].

Toll-like receptors (TLRs) knockout mice and cytokine-deficient mice both display the importance of TLRs for fungal recognition and cytokine production, such as TNF-alpha, in host defense [32,87,88,89]. Spikes et al. found that deletions in virulence factors, such as gliotoxin, show a decrease in virulence in non-neutropenic models while having no effect in neutropenic models [32,90]. *Aspergillus* endophthalmitis-specific models provide a means to investigate the pathogenesis of infections in both immunocompetent and immunocompromised hosts. Researchers have used these mice in studies examining the effect that specific drug treatments, such as isavuconazole and amphotericin B deoxycholate, have on treating the fungal infection, its toxicity, and its ability to restore vision back to baseline [22,91]. Different treatment routes can also be tested, as shown by Guest et al., where equal efficacy between the oral and intravitreal injection of isavuconazole in mice was observed [92].

Secondary metabolites released by *Aspergillus*, such as aflatoxin and gliotoxin, affect the contribution of neutrophils to host defense and inhibit the phagocytosis of fungal elements [87,93,94,95,96,97,98]. Dissemination of *Aspergillus*, typically from the lungs to other organs, is associated with neutropenia. *Aspergillus* spores enter the bloodstream, allowing it to spread and invade other tissues. After invasion, *Aspergillus* causes further damage, which in turn benefits and promotes fungal growth [93,99]. *Aspergillus* pathogenic factors have already been associated with specific immune responses. In mouse models of *Aspergillus* infection, researchers found an increase in neutrophil infiltration as well as an upregulation of TLRs (TLR1, 2, 3, 4, 6, 7, 8, and 9) with both pathogen-associated molecular patterns (PAMPs) and damage-associated molecular patterns (DAMPs) [91,100,101,102,103]. Previous reports suggest that TLRs expressed by retinal cells activate the retinal innate immune response, resulting in the production of inflammatory mediators (e.g., IFN-β, IL-6, IL-8, MCP-1, and ICAM-1) in response to pathogen invasion [91,100,101,102,103]. Shopova IA et al. demonstrated the production of human antifungal extracellular vesicles in response to *Aspergillus* infection in the lungs, which indicates an additional pathway the body utilizes to inhibit and kill the fungi. These vesicles could potentially represent a clinical marker for *Aspergillus* infection, thus directing further research to confirm the presence of these antifungal extracellular vesicles in the eye [104].

*Aspergillus* infection has been shown to cause a time-dependent increase in TLR (TLR2, 7, and 9) and pro-inflammatory mediator (IL-8 and TNF-α) expression in retinal pigment epithelial cells [105]. A significant up-regulation of IL-6, IL-10, IL-17, and MMP-9 was also seen in RPE cells infected with *A. flavus* [105]. Another study involving *A. flavus* and *C. albicans* infections in microglial cells (CHME-3) demonstrated that cells challenged with *A. flavus* expressed higher levels of TLRs (TLR-1, -2, -5, -6, -7 and -9) as well as cytokines (IL-1α, IL-6, IL-8, IL-10, and IL-17) in comparison to *Candida*. Both fungi up-regulated the expression of MMP-9 in the cultured microglia [106]. These reports indicate retinal residential cells evoke an inflammatory response via TLRs during fungal endophthalmitis. A study has also reported retinal disintegration and increased retinal cell death in *A. fumigatus* infected mouse eyes, with upregulation in TLR expression and inflammatory cytokines (IL-1β, IL-6 and TNF-α) and PMN infiltration [91]. Mice with neutropenia showed a higher disease severity with extended fungal burden and increased inflammatory mediators during exogenous *A. fumigatus* endophthalmitis [91]. Overall, a limited number of studies have assessed immune responses during *Aspergillus* endophthalmitis.

## 6. Diagnosis

Practitioners must diagnose fungal endophthalmitis as soon as possible to prevent possible vision loss. There is a wide range of tools available for the clinical diagnosis of fungal endophthalmitis. Typically, the initial examination is based upon the appearance of the eye, as well as any preexisting conditions or environmental factors which may aid in the subsequent diagnosis. After initial prognosis, testing of ocular fluids and tissues are carried out by different culture and molecular techniques [18,30].

### 6.1. Tissue Microscopy/Culture and Histopathology

Specimens collected from corneal scrapings, the conjunctival sac, and the vitreous or aqueous humor can be used for culturing in suitable media [18,63]. In exogenous fungal endophthalmitis, Gao et al. described approximately 50% of positive culture specimens as being isolated from corneal scrapings or corneal buttons, which corroborated with the initial location of *Aspergillus* seeding [107]. Additionally, the ocular tissues, such as corneal scrapings, can be better prepared with potassium hydroxide (KOH) and dimethyl sulfoxide (DMSO) digestion in KOH wet mounts [63]. This method allows tissue transparency for improved visualization and detection [108]. Culturing specimens can be inoculated on Sabouraud dextrose agar (SDA), blood agar, or chocolate agar, and incubated at both 37 °C and 27 °C for 7–10 days to monitor for any growth [63,109].

The aqueous and vitreous humors, which are typically obtained by pars plana vitrectomy (PPV) or a vitreous tap, are important specimens for the clinical diagnosis of endophthalmitis, as they can be used for either reverse transcription polymerase chain reaction (RT-PCR) or culturing [110]. It is pertinent to note that vitreous specimens are more reliable and sensitive than aqueous specimens in the process of fungal identification [111,112,113].

Although smears taken from vitreous specimens may be examined immediately via direct microscopy, they have a success rate of 50% for fungal detection. On the other hand, culturing is a common diagnostic method for fungi but has a relatively low sensitivity of 40%. This may be related to the slow growth rate or the required specific growing conditions of *Aspergillus*. Furthermore, cultures must be kept for at least 4–6 weeks to ensure that sluggish-growing or finicky fungal colonies are not missed [114]. Liu et al. used histopathological examinations of vitreous specimens to improve the detection rate to 70%. The combination of both methods, histology with PCR, can help achieve a more accurate diagnosis, suggesting its potency during initial fungal endophthalmitis speculation [114,115].

### 6.2. Blood Cultures

Although blood cultures can be used as a diagnostic tool, there is a large variation in positivity rates, from 33% to 50%, in all endophthalmitis cases [116,117]. Multiple case reports of endogenous endophthalmitis have reported false-negative cultures [17,21,26,68], while Zhang et al. reported positive blood cultures of *Aspergillus* panophthalmitis in a 10-year retrospective review of endogenous endophthalmitis [118]. However, without localized symptoms, diagnosis of *Aspergillus* endophthalmitis remains elusive [116,117].

### 6.3. Broad Range Real-Time PCR

To diagnose fungal endophthalmitis, quantitative real-time PCR is one of the best readily available options. When detecting a variety of fungal deoxyribonucleic acid (DNA), conserved sequences within the fungal genome are used to prepare primers and probes [119]. Sugita et al. utilized 18S ribosomal ribonucleic acid (rRNA) primers to detect six *Candida* and five *Aspergillus* species from either aqueous humor or vitreous fluid specimens [119]. In another study, Vollmer et al. targeted fungal 28S ribosomal DNA to detect *Candida, Aspergillus,* and *Cryptococcus* [119,120]. Real-time broad range PCR helps to rapidly identify the pathogenic fungus in comparison to the standard culturing method, which can speed up the treatment process [121]. Notably, the major drawbacks to PCR testing are its high cost and need for advanced equipment.

### 6.4. Imaging

To further aid in diagnosis, Adam et al. illustrated the use of enhanced depth imaging optical coherence tomography (OCT) to detect fungal endophthalmitis. Besides, wide-field fluorescein angiography (FA) can be used as evidence to rule out the presence of peripheral retinal vasculitis [65]. However, during disease progression, the fungus is no longer visible, which makes it difficult to detect, and hence an ultrasound examination can help in the process [18]. High-resolution computed tomography imaging (HRCT) and magnetic resonance imaging (MRI) can also be used to dynamically monitor the progression of the disease and aid in the following treatment [63].

### 6.5. (1→3)-β-D-Glucan Test

(1→3)-β-D-glucan (BDG), an endogenous polysaccharide found in the cell walls of certain fungi, is released into the bloodstream during invasive fungal infections [78]. This is commonly observed in systemic infections, but reports confirmed its presence in a few *Aspergillus* endophthalmitis cases [122]. An FDA-approved test for BDG detection is an enzyme-based colorimetric assay, commercially named Fungitell. The assay takes advantage of a modification of the Limulus amebocyte lysate for BDG quantification in the serum [122]. BDG has been extensively studied and validated during the early surveillance and diagnosis of fungal infections, particularly in patients with hematologic malignancy, a critical illness, and solid organs, and allogeneic hematopoietic stem cell transplant (HSCT) populations [122]. The guidelines of the Infectious Disease Society of America (IDSA) recommend estimating the levels of serum BDG for diagnosing invasive aspergillosis in high-risk patients such as those with hematologic malignancy and allogeneic HSC. Unfortunately, the assay is not specific for *Aspergillus* endophthalmitis [123] and hence appears to not be being used widely for the same [122]. A review article by Kolomeyer et al. showed increased serum BDG levels in fungal endophthalmitis, albeit only two studies showed elevated BDG levels in the vitreous. Due to this inconsistency, measurement of serum and vitreous BDG levels should be used more as an ancillary test to confirm a diagnosis, rather than as a primary tool. The study further suggested the use of BDG testing to monitor the progression of the disease and response to therapy [122].

### 6.6. Serum and Vitreous Galactomannan Assay

A major cell wall component of *Aspergillus* is the molecule galactomannan (GM), which is released at higher concentrations during *Aspergillus* infection [93]. A diagnostic immunoassay known as the Platelia GM enzyme assay currently identifies *Aspergillus*-specific samples [124]. With an accuracy of approximately 70%, studies showed the successful diagnosis of *Aspergillus* in a specific immunocompromised patient cohort, suffering either from hematological malignancy or having undergone allogeneic HSCT [125,126,127]. However, increased GM levels in the serum during conditions of neutropenia can affect antifungal efficiency [123]. Additionally, a major issue concerning the use of GM as a diagnostic factor is the lack of a standard range of detection for immunocompromised patients. Nevertheless, some cases involving *Aspergillus* endophthalmitis patients showed elevated levels of serum GM, and this thus might assist in the prognosis of *Aspergillus* infections [17,30]. Dupont et al. suggested that antigen testing of GM in the vitreous fluid could be a helpful tool in cases where PCR is unavailable, or mycology is negative [128]. Similarly, a more recent study reported that vitreous Galactomannan (GM) and 1,3 β-D-Glucan (BDG) levels can be used for the diagnosis of fungal endophthalmitis in culture-negative cases [129].

## 7. Antifungal Management

Due to the relative scarcity of *Aspergillus*, the exploration of usable drug types has been limited to three classes of antimycotics as primary modes of treatment. In a clinical setting, these drugs belong to the families of triazoles, echinocandins, or polyenes, and they act by disrupting the integrity of the *Aspergillus* cell wall [125]. According to Infectious Diseases Society of America (IDSA) guidelines, drugs for fungal endophthalmitis are administered via intravitreal injections or intravenous routes [82]. Typically, intravenous amphotericin B (Amb) or voriconazole and oral flucytosine are the preferred agents with a vitrectomy, whereas severe vision-threatening conditions are treated with intravitreal Amb. In invasive *Aspergillus* infection, successful treatment has also been achieved with the addition of triazoles or polyenes, although fluconazole is used in less severe cases [14,125]. The overall rarity of *Aspergillus* endophthalmitis has resulted in a lack of clinical trials. Thus, it is crucial to consider all forms of treatment to better serve patients’ interests and their overall vision.

### 7.1. Polyenes

Polyenes are a versatile group of anti-mycotic drugs, one of the most widely used drugs in *Aspergillus* endophthalmitis treatment, and include amphotericin B (AmB) [11,17,21,61,65]. Amphotericin B binds to ergosterol and prevents *Aspergillus* from having a functional cell wall. Amphotericin B is composed of two different molecular domains: a hydrophilic polyhydroxy chain and a hydrophobic polyene hydrocarbon chain. These two domains are critical for their ability to interact with ergosterol and thus their capacity to assert antifungal effects [126]. Amphotericin B deoxycholate poses a major hurdle as a systemic treatment option: its dose-dependent toxicity, and more specifically, its nephrotoxicity [126,127]. To attenuate the substantial toxicity of amphotericin B while also improving its therapeutic capacity, three different lipid formulations have been developed: an amphotericin B colloidal dispersion (ABCD), liposomal amphotericin B (L-AmB), and amphotericin B–lipid complex (ABLC) [126]. It is crucial to attain the necessary concentration of medicine in the affected tissues to successfully treat fungal infections, which can become challenging in poorly vascularized tissues [82]. A major issue in *Aspergillus* endophthalmitis treatment is that the systemic intravenous (IV) administration of AmB-d will not ensure the necessary therapeutic levels within the eye [91]. Moreover, to overcome this issue, injections of AmB-d can be administered directly into the vitreous [82]. The dose-dependent toxicity of the retina was observed in rabbits who were given intravitreal injections of AmB-d by Axelrod et al. [130]. Dose-related toxicity has been observed with all forms of AmB, although liposomal AmB is less toxic than either AmB lipid complex or AmB-d [82,131]. Unfortunately, a lack of definitive and decisive information on the ultimate effectiveness of intravitreal injections of AmB-d persists. Reports have shown a wide range of doses, ranging from 20 to 100 μg with non-cytotoxic effects, but it should be noted that administered doses typically range from 5 to 10 μg [82]. In an evaluation of 4 patients, Bae et al. reviewed 7 eyes that were treated with intravitreal liposomal AmB, which was found to be safer and better tolerated than AmB-d. [132]. In cases of macular infection or vitritis from *Aspergillus* endophthalmitis, a combination of systemic and intravitreal antifungal agents should be used to ensure appropriate therapeutic levels within the posterior segment of the eye. To achieve this, repeated injections may be necessary, but they should only be performed after a thorough examination after the first injection [82]. Combination therapy is outlined by IDSA, which suggests *Aspergillus* endophthalmitis to be treated with either intravitreal AmB deoxycholate or voriconazole with oral or intravenous voriconazole [123]. In a separate study, most patients with endogenous endophthalmitis had partially or fully recovered from ocular lesions with the systemic administration of AmB alone or in combination with an oral antifungal, whereas those with exogenous endophthalmitis suffered permanent blindness, enucleation, and evisceration [74].

### 7.2. Azoles (Triazoles)

Triazoles are known drugs to inhibit the development of fungal cell membranes. They work by inhibiting the ability of *Aspergillus* to synthesize ergosterol via a blockade of 14α-demethylase [125]. Of note, the popular agent fluconazole can be used against *Candida*, yet it is ineffective in the treatment of mold [76].

*a*.
*Voriconazole*


A widely used agent for the treatment of various fungal infections is voriconazole (134). The preferred treatment involves direct injection of voriconazole into the vitreous to effectively achieve therapeutic levels. Similar to other systemic antifungal agents, the necessary drug concentration is not feasible when given orally or via an intravenous route and has poor ocular penetration [82]. In vitro studies have shown *Aspergillus* requires a minimum inhibitory concentration (MIC) for voriconazole of at least 0.5 μg/mL [133]. Dave et al. performed a retrospective study of 91 eyes in 91 patients who had *Aspergillus* endophthalmitis and found that those treated with intravitreal voriconazole resulted in a favorable visual outcome [13]. Moreover, voriconazole can be used in combination or alone with AmB-d when given as intravitreal injections. With the advent of voriconazole, research suggests it is a safer option than AmB-d; however, as AmB-d has been in use for many years and has a longer half-life after intravitreal injection administration, the benefits of voriconazole still need to be studied. Due to voriconazole’s shorter half-life, there may be a need for repeated injections depending on the response to therapy after examination [82,134]. Mithal et al. suggested a combination regimen, consisting of daily intravitreal voriconazole injections and alternate-day amphotericin B in cases of persistent fungal endophthalmitis [13,135,136].

Another limitation of voriconazole is the possibility of drug interactions that are common with azole drug use. It is important to specify that many of the following observations are associated with intravenous or oral administration and may not be relevant in intravitreal injections [125]. Nevertheless, deleterious effects should still be considered when choosing appropriate treatment options. Specifically, azole drugs interact with both CYP3A4 and 2C19, in which the P450 enzymes are responsible for metabolizing various drugs. Due to azole drugs interacting with the P450 enzymes, the proton-pump inhibitors, HIV protease inhibitors, cyclosporine, HMG-CoA reductase inhibitors, corticosteroids, and warfarin may be metabolized at a slower rate. Patients with heart disease should be cautioned when azoles are used in combination with quinolones, as this can lead to prolongation of the QTc interval. Contraindications have occurred when voriconazole has been used in combination with the immunosuppressive agent sirolimus or the antiviral efavirenz. Another classification of drugs that should be modified includes calcineurin inhibitors. According to Panackal et al., when calcineurin inhibitors are used in conjunction with voriconazole or other azole drugs, a reduction in the dosage of anywhere from 25 to 75% should be considered. Due to the interactions of azole drugs with the liver, the overall organ function should also be monitored due to possible increases in liver enzymes [125].

Zhao et al. performed a comparative study between intravitreal voriconazole and liposomal AmB in Guinea pigs with *Aspergillus* endophthalmitis. From a treatment point of view, liposomal AmB and intravitreal voriconazole have been shown as promising antifungal compounds. This study also showed voriconazole to be more effective against acute *Aspergillus* infections in comparison to liposomal AmB, with neither having retinal toxicity at an intravitreal dose of 20 μg/0.02 mL [137,138].

*b*.
*Isavuconazole*


Isavuconazole is a second-generation triazole antifungal agent that can be used in a variety of topical and oral applications, including the treatment of aspergillosis, *Candida*, and mucormycosis [139,140]. Isavuconazole also works by inhibiting the development of the cell membrane via the inhibition of ergosterol [140]. There are a few major advantages in comparison to voriconazole when used in invasive aspergillosis—it has a longer half-life, 56–104 h, without reported renal or phototoxicity, no adverse central nervous system effect, and broader antifungal coverage outside of *Aspergillus* and *Candida* [32,141,142]. A major advantage of isavuconazole over other azole drugs is its lower rate of inhibition with cytochrome P450, thus leading to fewer drug interactions [32].

Although there are no clinical studies that have successfully shown isavuconazole as a form of treatment for endophthalmitis in humans, a recent study by Guest et al. showed efficacy in the treatment of *A. fumigatus* endophthalmitis in C57BL/6J mice by isavuconazole. Isavuconazole was administered by oral gavage, intravenous, intravitreal, and oral/intravitreal and intravenous/intravitreal routes in combination. The results of this study showed oral administration was as effective as an intravitreal injection alone and has laid the groundwork for a possible alternative in treating *Aspergillus* endophthalmitis. In their mouse model, isavuconazole reduced the fungal burden, helped retain the retinal structure and function, and inhibited inflammatory cytokine production [90]. This proof-of-concept study demonstrates the ability of isavuconazole for the treatment of Aspergillus endophthalmitis, although studies are lacking in human.

### 7.3. Echinocandins

Another group of anti-fungal agents that has gained notoriety as a treatment option for *Aspergillus* endophthalmitis is the echinocandins. Like other antimycotics, echinocandins work by inhibiting the synthesis of the fungal cell membrane. More specifically, it non-competitively inhibits 1,3-β-d glucan synthase, which is necessary to produce BDG [143]. Previous studies have shown that the inadequate penetration of echinocandins into various ocular compartments inhibits its ability to reach therapeutic levels, and direct administration is therefore advised [82]. A drawback of caspofungin, amongst other echinocandins, is its inability to penetrate the eye when given systemically [22,86,96,135,143]. This is due to a combination of its large molecular weight and the strong protein binding properties it possesses, with previous attempts of administering caspofungin intravenously failing to treat the eye [22,86,96,135,143]. Case reports have shown that a combination of intravenous caspofungin and intravitreal voriconazole has resulted in the improvement of endophthalmitis in patients [87,136].

To determine the overall safety and pharmacokinetics, Shen et al. performed a study of intravitreally injected caspofungin in rabbits [133]. Rabbit eyes injected with concentrations of 50 μg/0.1 mL of caspofungin showed no sign of toxicity in their eye and no differences in either their *a*-wave or *b*-wave responses [133]. Another study by Mojumder et al. evaluated the retinal toxicity of intravitreal caspofungin in mice. Mice were injected with caspofungin at various concentrations and examined by both electroretinography (ERG) and histopathology. Only eyes treated with 41 μM and higher concentrations had deleterious effects. According to Mojumder *et al*., this translates to a threshold limit of 20 μg/4 mL [143,144].

Additionally, a study by Kernt et al. investigated the in vitro toxicity of caspofungin in different ocular cell lines. Doses between 5 and 300μg/mL were used to determine the effects on corneal endothelial cells (CEC), endothelial cells, primary human trabecular meshwork cells (TMC), and primary human retinal pigment epithelium (RPE) cells. Cellular toxicity was not observed in any of the cell types used till 50 μg/mL [88,145]. Despite many studies showing cases of safe use on patients, echinocandins are no longer recommended by IDSA guidelines for endophthalmitis treatment.

### 7.4. Combination Therapy

Mithal et al. performed a retrospective, non-randomized study of 12 eyes diagnosed with fungal endophthalmitis. Patients were treated with a combination of daily intravitreal AmB 5 µg/0.1 mL and intravitreal voriconazole 100 µg/0.1 mL combined with a PPV. The results of the study looked promising, with a corrected visual acuity of 20/400 or better in 7 of the 12 eyes and 20/60 in 2 eyes. There was no loss in light perception and no enucleation was necessary [141].

### 7.5. Corticosteroids

Inflammation elicited by infection can result in the overactivation of host immune responses, destroying various affected tissues [146]. One mode of combating inflammation, especially in cases of severe inflammation, is the use of corticosteroids [142]. Studies suggest the use of intravitreal dexamethasone or oral prednisone (1 mg per kg of body weight) for the treatment of endophthalmitis [18]. However, the use of corticosteroids has a few major drawbacks, including inhibition of anti-fungal agents, and therefore proper precautions should be taken if used in conjunction with other therapies. Long-term systemic use of corticosteroids creates a risk factor for the development of invasive fungal infections. Due to the paradoxical roles of corticosteroids in both combating excess inflammation as well as inhibiting the immune response, their use in patients as a treatment intravitreally has been controversial [147,148]. A retrospective study by Majji et al. showed patients who were treated with dexamethasone had better visual acuity in comparison to those who did not receive the treatment. The results hint that proper application of the corticosteroids could result in beneficial effects, though other factors, such as the duration, should be considered [148,149].

## 8. Surgical Management

### 8.1. Vitreous Tap vs. Pars Plana Vitrectomy

Vitrectomies have been used for both diagnostic and therapeutic purposes in retinal diseases. Regarding diagnosis, roughly 90% of cultured vitrectomy specimens are positive [122]. In contrast, only 60% of vitreous aspirates are positive [59]. Although PPVs can be used for retinal reattachment, under specific circumstances, the opposite can occur due to excessive force, resulting in retinal detachment [122].

Multiple reports show early intervention with a vitrectomy could result in a better visual outcome when treating fungal endophthalmitis [150,151]. Behera et al. analyzed outcomes of 66 patients where *Aspergillus* was the most common pathogen isolated. In cases where an early vitrectomy plus intravitreal antifungals was performed, a more favorable outcome was observed in comparison to only having a diagnostic vitrectomy with intravitreal antifungals [64]. A study by Dave et al. showed that those who received a PPV had both improved visual and anatomical outcomes than those who only received a vitreous tap [13]. In the case of fungal keratitis, Gao et al. suggested a vitrectomy as a preventative measure against endophthalmitis, especially in situations where intravitreal injections fail to work or in more advanced cases of fungal keratitis [107].

### 8.2. Evisceration

Typically, evisceration is reserved as a last-resort treatment choice against endophthalmitis when the infection cannot be controlled. According to Dave et al., evisceration or enucleation can climb as high as 25% once fungi become filamentous [11,152]. A retrospective study of infectious endophthalmitis that resulted in evisceration showed *Aspergillus* accounting for 58 cases, with *Streptococcus pneumoniae* being the one more common causative agent, accounting for 68 cases out of 388 [152].

## 9. Conclusions

Fungal infections of the eye that go untreated often lead to blindness. *Aspergillus* endophthalmitis can spread through either endogenous or exogenous routes, and most commonly infects immunocompromised patients due to post-surgery ocular trauma, organ transplantation, or ocular trauma. Advanced diagnostic tools such as fluor angiography, fluorescence in situ hybridization (FISH), and retinochoroidal biopsy can potentially aid in complicated ocular infections [18]. A comprehensive understanding of various treatment options and their effectiveness has emerged as the overarching theme for potential areas of future research. Given the emerging drug-resistant strains and fewer available anti-fungal treatment options, studies on effective combination therapy should be considered [29,153]. Flucytosine can be used as an adjunct therapy in combination with AmB. Additionally, oral and topical administration of Posaconazole might be considered as a potential antifungal for endophthalmitis [82]. Due to the perniciousness of this disease, it may be advantageous to administer prophylactic drugs in higher-risk situations before scheduled surgeries to prevent infections and final vision loss, although studies are needed to support this claim [59].

Currently, most animal models of *Aspergillus* endophthalmitis have been used to test the therapeutic efficacy of antifungal drugs. Although potentially blinding, little research has focused on understanding the basic pathobiology of *Aspergillus* endophthalmitis. Similarly, more investigations are needed for laboratory diagnosis of *Aspergillus* endophthalmitis, including the use of omics (transcriptomic, proteomic and metabolomics) to identify potential disease biomarkers.

## Figures and Tables

**Figure 1 jof-08-00656-f001:**
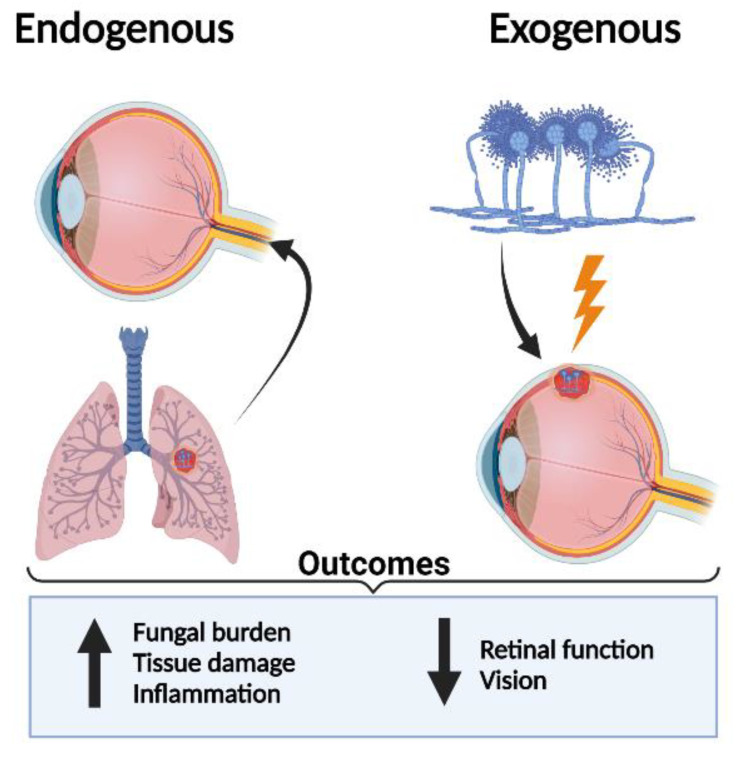
Fungal spores are inhaled from the environment into the lung resulting in lung Aspergillosis. As the disease progresses, spores are released into the bloodstream and disseminate to other organs, including the eye, resulting in endogenous *Aspergillus* endophthalmitis. Fungal hyphae eventually cross the blood–retinal barrier and penetrate inside the eye. In the case of exogenous endophthalmitis, *Aspergillus* spores enter the eye during ocular trauma.

**Figure 2 jof-08-00656-f002:**
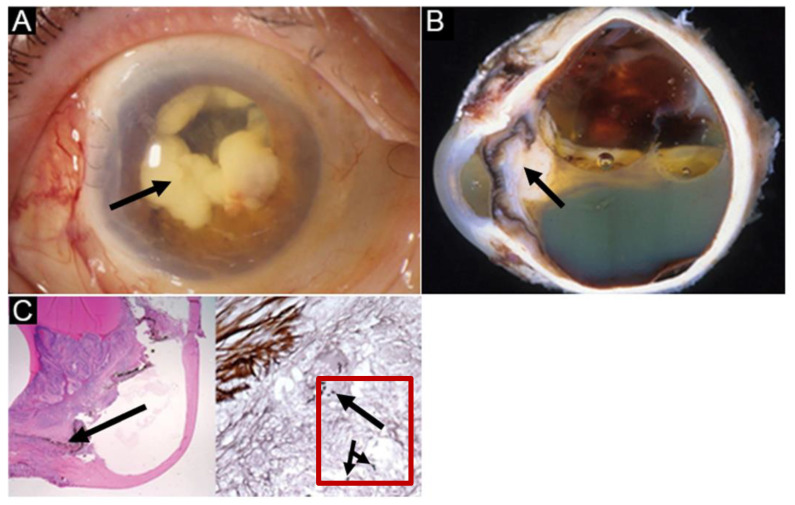
(**A**) An image of a 69-year-old woman’s infected eye 2 months post-cataract surgery. The eye illustrates extensive *Aspergillus* infiltration in the anterior chamber. *Aspergillus* endophthalmitis treatment includes multiple procedures: a pars plana vitrectomy, lens removal, and amphotericin B injections. Due to a poor response to mainstream treatment options, enucleation was performed as a last resort. (**B**) A cross-section image of the eye after enucleation. The voracity of the infection is apparent with a clear detachment of the retina. The intense creamy white plaque around the iris displays the virulent spread of *Aspergillus*. (**C**) An H&E-stained histopathological image of the infected eye with higher magnification is shown on the right. The image on the right reveals individual *Aspergillus* hyphae formation, shown in the red box with corresponding arrows that occurs during infection spread. Image courtesy: Haddock et al. [62].

**Figure 3 jof-08-00656-f003:**
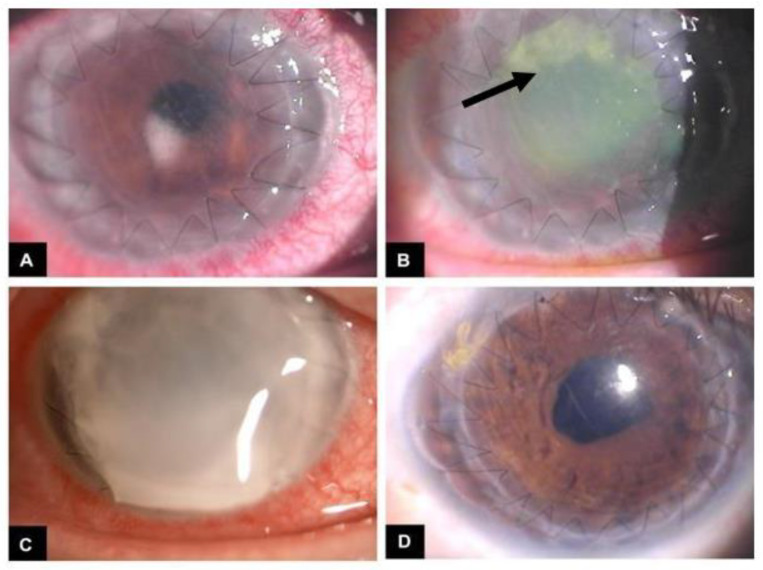
A biomicroscopic image of the disease progression and treatment of an *Aspergillus* infection after penetrating keratoplasty (PKP). (**A**) Eye 4 days after PKP. (**B**) After 6 days, the progression of the disease is quite apparent with white infiltrates and stromal melting (arrow). (**C**) Nine days after initial infection, with the anterior chamber being affected completely. (**D**) Three months later, showing clearance of the disease with restoration in the clarity of the anterior chamber and corneal graft. Image courtesy: Spadea et al. [18].

## Data Availability

Not applicable.

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
