# Peer review of "Aspergillus* Endophthalmitis: Epidemiology, Pathobiology, and Current Treatments"

_jof, 2022, doi:10.3390/jof8070656_

Round 1

Reviewer 1 Report

The manuscript #jof-1720270, entitled “Aspergillus Endophthalmitis: Epidemiology, Pathobiology, and Current Treatments” by Khambati et al. presents a comprehensive review on, in line with the title, endophthalmitis caused by Aspergillus spp., with the main focus on A. fumigatus. The manuscript is well planned and consists of the following sections: epidemiology, description of the most common etiological factors, clinical manifestations, immune response towards the infection, diagnosis routes and treatment options. What is worth mentioning is the fact that all those sections were described keeping in mind the main theme of the manuscript - eye infections. My only issue is the fact that the manuscript requires some edition - examples are below:

  • numerous examples of using "Aspergillus" or "A. fumigatus" unitalicized;
  • line 32: broken sentence;
  • line 192: unnecessary space before the bracket. 

Author Response

numerous examples of using "Aspergillus" or "A. fumigatus" unitalicized;

Response: We have changed the scientific names to italics throughout the manuscript.

  • line 32: broken sentence

Response: We revised the referred sentence.

Reviewer 2 Report

The authors give an overview about aspergillus endophtalmitis. The review is relatively long and much is about general aspergillus infections. For better readability, the authors should probably focus more specifically on endophtalmitis and reduce the general sections about aspergillosis and its immunology.  

Major comments 

1. The section 3 maybe is redundant. The first part of section 4 is repeating the previous sections, and the sections about the different Aspergillus species is not very informative for the specific subject. Is there anything spesific know about this pathogen in relation to endophtalmitis? if not, probably delete (or reduce) this whole section 

2. Section 5 nicely summeries the immune response in aspergillus infections. However, only 1 or 2 lines are about endophtalmitis specific. I suggest to end with an paragraph/ overview about what is know about the immune response related to aspergillus endophtalmitits 

3. Sections on antifungals can be reduced by focussing on endophtalmitis only. Probably also mention the ocular penetration of iv use of the agents. . 

Section 4 is the most important section. Any difference the presentation or ocular signs in in endogenous/exogenous aspergillus endophtalmitis? 

Minor comments. 

Line 18, remove (A). 

Line 29, flavus

Line 31, only in unhygienic home environments?

line 34, A. fumigatus exposure?

line 37, immunocompromised patients dont develop Aspergilloma. Patients with cavities develop aspergilloma. 

Line 39 dissemination is relativey rare 

line 45, include refs for the prevalences

Line 74: the incidence of endogenious fungal endophtalmitis. Incidence in which patients? patients with invasive fungal infections? please clarify, For The incidence of Aspergillus endophtalmitis in invasive aspergillosis is likely much lower than reported 2 to 40 %. 

Line 111. what cellular heterogeneity?

Line 330-331. 33-94% in endogenous bacterial infections i presume. probably >0.1% in aspergilllus infections. Bloodcultures for aspergillus infections are useless.   

395-397 rephrase sentence. 

424-425 also the prefered treatment for endophtalmitis? 

511 drawback of caspofungin only or all echinocandins? 

516-517 synergy in aspergillus endophtalmitis? 

7.5 line 539: what do guidelines advise about corticosteroid use in fungal endophtalmitis. 

Author Response

The section 3 maybe is redundant. The first part of section 4 is repeating the previous sections, and the sections about the different Aspergillus species is not very informative for the specific subject. Is there anything spesific know about this pathogen in relation to endophtalmitis? if not, probably delete (or reduce) this whole section.

Response: We have deleted redundant information in section 3 and added new information based on endophthalmitis in different Aspergillus species.

Section 5 nicely summeries the immune response in aspergillus infections. However, only 1 or 2 lines are about endophtalmitis specific. I suggest to end with an paragraph/ overview about what is know about the immune response related to aspergillus endophtalmitits 

Response: We have added additional details in a separate paragraph regarding the immune response in Aspergillus-induced endophthlamitis.

Sections on antifungals can be reduced by focussing on endophtalmitis only. Probably also mention the ocular penetration of iv use of the agents.

Response: We have reduced extraneous information and focused more into endophthalmitis tratment. New sentences related to IV penetration have also been added as suggested.

Section 4 is the most important section. Any difference the presentation or ocular signs in in endogenous/exogenous aspergillus endophtalmitis?

Response: We have added information regarding clinical presentations/outcomes for endogenous and exogenous endophthalmitis in both section 4 and section 6.

Line 18, remove (A). 

Response: We have removed the (A)

Line 29, flavus

Response: We have modified the sentence.

Line 31, only in unhygienic home environments?

Response: We remove the word “unhygienic” as they can found in any home especially within damp and shady areas.

line 34, A. fumigatus exposure?

Response: We changed the sentence to represent its exposure.

line 37, immunocompromised patients dont develop Aspergilloma. Patients with cavities develop aspergilloma. 

Response: We have rephrased the sentence to specify that patients with cavities can develop Aspergilloma.

Line 39 dissemination is relativey rare 

Response: We have added that dissemination rarely happens

line 45, include refs for the prevalences

Response: We have added the reference

Line 74: the incidence of endogenious fungal endophtalmitis. Incidence in which patients? patients with invasive fungal infections? please clarify, For The incidence of Aspergillus endophtalmitis in invasive aspergillosis is likely much lower than reported 2 to 40 %. 

Response: We revised the statement.

Line 111. what cellular heterogeneity?

Response: We have clarified what cellular heterogeneity refers to.

Line 330-331. 33-94% in endogenous bacterial infections i presume. probably >0.1% in aspergilllus infections. Bloodcultures for aspergillus infections are useless.   

Response: We have added that 33-94% refers to endogenous bacterial infections.

Line 395-397 rephrase sentence. 

Response: We have rephrased the sentence.

Line 424-425 also the prefered treatment for endophtalmitis? 

Response: We have deleted the sentence specifying only liposomal form is preferred treatment.

Line 511 drawback of caspofungin only or all echinocandins? 

Response: We have clarified the sentence.

Line 516-517 synergy in aspergillus endophtalmitis? 

Response: We have reworded the synergy to reflect combination therapy.

line 539: what do guidelines advise about corticosteroid use in fungal endophtalmitis.

Response: We provided the information.

Reviewer 3 Report

In the manuscript entitled "Aspergillus Endophthalmitis: Epidemiology, Pathobiology, and Current Treatments”.

The authors have described the various aspects of Aspergillus Endophthalmitis. This manuscript is interesting and can be educational. By the way, there are some points that should be corrected. Because although it has very new content, some sentences have used old references that need to be revised and some sentences are without references!

  1. Lines 18, 22, 26, ..: “Aspergillus” should be Italic inside of all text. “Aspergillus (A) fumigatus” to be “ fumigatus”.
  2. Lines 28-29, it is better to omit it. It is not suitable for this manuscript.
  3. Line 30, “The genus typically lives ….” To be “The Aspergillus genus typically lives ….”.
  4. Line 32: First, Reference 1 is too old! You can use a new one too, and suggested reference “ High Prevalence of Clinical and Environmental Triazole Resistant Aspergillus fumigatus in Iran: Is It a Challenging Issue? J Med Microbiol. 2016; 65: 468-475.” about A. fumigatus importance. Second, in some area this species is not prevalent and you should revise this sentence “Although A. fumigatus isreported as the most prevalent species responsible for 90% of human infections, non-fumigatus species are increasing due to various reasons.” and add a new related reference too, and suggested reference “b. Predominance of non-fumigatus Aspergillus species among patients suspected to pulmonary aspergillosis in a tropical and subtropical region of the Middle East. Microbial Pathogenesis. 2018; 116: 296–300.”.
  5. Line 37, “…growth in the lungs.” This sentence needs some references too, and suggested references “ Detection of Aspergillus flavus and A. fumigatus in Bronchoalveolar Lavage Specimens of Hematopoietic Stem Cell Transplants and Hematological Malignancies Patients by Real-Time Polymerase Chain Reaction, Nested PCR and Mycological Assays. Jundishapur J Microbiol. 2015; 8(1): e13744. d. Mycological Microscopic and Culture Examination of 400 Bronchoalveolar Lavage (BAL) Samples. Iranian J Publ Health. 2012; 41(7): 70-76.”.
  6. Line 44, “et al.” should be Italic inside of all text.
  7. Figure 1, ERG need an explanation!
  8. Line 93, “…primarily as opportunistic pathogens in humans.” This sentence needs reference too, and suggested references “ Assessment of indoor and outdoor airborne fungi in an Educational, Research and Treatment Center. Italian Journal of Medicine. 2017; 11: 52-56.”.
  9. Line 103, “…centrations and continuously inhaled.” This sentence needs reference too, and suggested reference “”.
  10. Line 115, “…due to compromised fungicidal ability.” This sentence needs reference too, and suggested reference “ Successful treatment of pulmonary aspergillosis due to Aspergillus fumigatusin a child affected by systemic lupus erythematosus: A case report from Northeastern Iran. Clinical Case Reports. 2021;9(5):1–9. e04248.  ”.
  11. Line 131, “…with non-CGD patients as well..” This sentence needs reference too, and suggested reference “g. Genetic diversity and antifungal susceptibility patterns of Aspergillus nidulans complex obtained from clinical and environmental sources. 2020; 63: 78-88.”.
  12. Line 141, references 50 and 52 are old for this interesting manuscript!
  13. Figure 2, A and B, The places involved in the photo should be marked with an arrow. The C section needs more magnification and the type of tissue staining should be mentioned too.
  14. Figure 3, The places involved in the photo should be marked with an arrow.
  15. Line 305, please use “specimen” instead of “sample” inside of all text.
  16. Line 345, “…in comparison to the standard culturing method, which can speed-up the treatment process.” This sentence needs reference too, and the suggested reference “h. Use of mycological, nested-PCR , and real-time PCR methods on BAL fluids for detection of Aspergillus fumigatus and flavus in solid organ transplant recipients. Mycopathologia. 2013; 176(5-6): 377-385.”.
  17. Line 380, “…GM enzyme assay currently identifies Aspergillus-specific samples..” This sentence needs reference too, and suggested reference “k. Effect of involved Aspergillusspecies on galactomannan in bronchoalveolar lavage of patients with invasive aspergillosis. J Med Microbiol. 2017; 66: 898-904.”.
  18. The conclusion section needs to mention better laboratory diagnostic methods and more effective antifungals.

Author Response

Lines 18, 22, 26, ..: “Aspergillus” should be Italic inside of all text. “Aspergillus (A) fumigatus” to be “ fumigatus

Response: We have changed the scientific names to italics throughout the manuscript.

Lines 28-29, it is better to omit it. It is not suitable for this manuscript

Response: We have modified the sentence accordingly.

Line 30, “The genus typically lives ….” To be “The Aspergillus genus typically lives ….”

Response: We have added ‘Aspergillus’ as suggested.

Line 32: First, Reference 1 is too old! You can use a new one too, and suggested reference “ High Prevalence of Clinical and Environmental Triazole Resistant Aspergillus fumigatus in Iran: Is It a Challenging Issue? J Med Microbiol. 2016; 65: 468-475.” about A. fumigatus importance. Second, in some area this species is not prevalent and you should revise this sentence “Although A. fumigatus isreported as the most prevalent species responsible for 90% of human infections, non-fumigatus species are increasing due to various reasons.” and add a new related reference too, and suggested reference “b. Predominance of non-fumigatus Aspergillus species among patients suspected to pulmonary aspergillosis in a tropical and subtropical region of the Middle East. Microbial Pathogenesis. 2018; 116: 296–300.”.

Response: We have changed reference 1 and cited new reference 5 and revised the statement.

Line 37, “…growth in the lungs.” This sentence needs some references too, and suggested references “ Detection of Aspergillus flavus and A. fumigatus in Bronchoalveolar Lavage Specimens of Hematopoietic Stem Cell Transplants and Hematological Malignancies Patients by Real-Time Polymerase Chain Reaction, Nested PCR and Mycological Assays. Jundishapur J Microbiol. 2015; 8(1): e13744. d. Mycological Microscopic and Culture Examination of 400 Bronchoalveolar Lavage (BAL) Samples. Iranian J Publ Health. 2012; 41(7): 70-76.”.

Response: We have included the related references as per suggestion.
Line 44, “et al.” should be Italic inside of all text.

Response: The correction has been done in the revised manuscript.
Figure 1, ERG need an explanation!

Response: ERG is used to assess retinal function. However, for broad audience we now removed the word ERG from Fig.1 and mentioned reduction in  “retinal function” and “vision”.

Line 93, “…primarily as opportunistic pathogens in humans.” This sentence needs reference too, and suggested references “Assessment of indoor and outdoor airborne fungi in an Educational, Research and Treatment Center. Italian Journal of Medicine. 2017; 11: 52-56.”.

Response: We have cited the suggested reference.

Line 103, “…centrations and continuously inhaled.” This sentence needs reference too, and suggested reference “”.

Response: We have provided the citation.

Line 115, “…due to compromised fungicidal ability.” This sentence needs reference too, and suggested reference “Successful treatment of pulmonary aspergillosis due to Aspergillus fumigatusin a child affected by systemic lupus erythematosus: A case report from Northeastern Iran. Clinical Case Reports. 2021;9(5):1–9. e04248.”

Response: We have added the suggested reference.

Line 131, “…with non-CGD patients as well.” This sentence needs reference too, and suggested reference “g. Genetic diversity and antifungal susceptibility patterns of Aspergillus nidulans complex obtained from clinical and environmental sources. 2020; 63: 78-88.”

Response: We have added the suggested reference.

Line 141, references 50 and 52 are old for this interesting manuscript

Response: We have kept the references as they are important for the review and the statement written. There are newer references mentioned alongside as well.

Figure 2, A and B, The places involved in the photo should be marked with an arrow. The C section needs more magnification, and the type of tissue staining should be mentioned too.

Response: We have added arrows in the images for better understanding. One magnified section has been added for the C section and the type of tissue staining has been mentioned in the figure legend.

Figure 3, The places involved in the photo should be marked with an arrow.

Response: We have added arrows in the images for better understanding.

Line 305, please use “specimen” instead of “sample” inside of all text.

Response: We have made the suggested changes in the revised manuscript.

Line 345, “…in comparison to the standard culturing method, which can speed-up the treatment process.” This sentence needs reference too, and the suggested reference “h. Use of mycological, nested-PCR , and real-time PCR methods on BAL fluids for detection of Aspergillus fumigatus and flavus in solid organ transplant recipients. Mycopathologia. 2013; 176(5-6): 377-385.”

 Response: We have added a new reference to support the statement.

Line 380, “…GM enzyme assay currently identifies Aspergillus-specific samples..” This sentence needs reference too, and suggested reference “k. Effect of involved Aspergillusspecies on galactomannan in bronchoalveolar lavage of patients with invasive aspergillosis. J Med Microbiol. 2017; 66: 898-904.”.

Response: We have added the suggested reference to support the statement.

The conclusion section needs to mention better laboratory diagnostic methods and more effective antifungals.

Response: We have mentioned few diagnostic tools and antifungals to improve the manuscript.